# Novel Biomarkers and Therapeutic Targets for Melanoma

**DOI:** 10.3390/ijms231911656

**Published:** 2022-10-01

**Authors:** Noa Sabag, Alexander Yakobson, Meir Retchkiman, Eldad Silberstein

**Affiliations:** 1Department of Plastic Surgery, Soroka University Medical Center, Ben-Gurion University of the Negev, P.O. Box 151, Beer Sheva 8410100, Israel; 2Oncology Institute, Soroka University Medical Center, Faculty of Health Sciences, Ben-Gurion University of the Negev, P.O. Box 151, Beer Sheva 84100, Israel

**Keywords:** malignant melanoma, immunotherapy, adjuvant therapy, targeted therapy

## Abstract

Malignant melanoma is one of the most common cancers in the world. In the disease’s early stages, treatment involves surgery, in advanced stages however, treatment options were once scarce. There has been a paradigm shift in advanced melanoma treatment with the introduction of immunotherapy and targeted therapies. Understanding the molecular pathways and their pathologic counterparts helped identifying specific biomarkers that lead to the development of specific targeted therapies. In this review we briefly present some of these markers and their relevance to melanoma treatment.

## 1. Introduction

Malignant melanoma is one of the most common cancers in the world, ranking fifth for the highest number of estimated new cases in 2022 according to Cancer Statistics Cases [1]. In the disease’s early stages, treatment involves surgery; in advanced stages, however, treatment options were once scarce. There has been a paradigm shift in advanced melanoma treatment with the introduction of immunotherapy and targeted therapies. This began in 2011 when the U.S. Food and Drug Administration (FDA) approved the first drug, Ipilimumab, that proved to decrease mortality rates [2]. Since then, the FDA has approved eight more drugs for the treatment of melanoma that have resulted in improved overall survival rates [3].

Many patients, however, develop either primary or even secondary resistance to current systemic treatment options. Resistance is due to multiple complex mechanisms driven by a set of rewiring processes that involve cancer metabolism, epigenetics, gene expression and interactions in the tumour microenvironment.

There are several molecular pathways that, when unregulated, may become pathological and influence the onset, proliferation and invasion of melanoma. 

The five main pathways are: The Mitogen-activated Protein Kinase (MAPK) pathway, where serine/threonine-protein kinase BRAF (BRAF) is involved [4];The phosphatidylinositol-3-kinases (PI3K)-AKT pathway [5];The gene suppressor Cyclin-Dependent Kinase Inhibitor 2A (CDKN2A) and tumour suppressors pathway [6];The microphthalmia-associated transcription factor (MITF) pathway [7];The nuclear factor-kappa B (NFκB) pathway [8].

Understanding these molecular pathways and their pathologic counterparts helps identify specific biomarkers that lead to the development of specific targeted therapies.

We briefly review some of these markers and their relevance to melanoma treatment.

## 2. Malignant Melanoma Classification

The classical prognostic markers regarded as the “Gold Standard” for melanoma treatment are histological: Breslow thickness, Clark level ulceration and mitotic rate [9]. By utilizing these markers, the ‘T’ of the TNM classification is established, and the prognosis is estimated. However, in addition to histopathological findings, clinical characteristics are essential, as melanomas may develop into one of two main pathways. The first pathway is associated with melanocyte proliferation, and the second with chronic exposure to sunlight [10].

Specific mutations may correlate with sun exposure. BRAF mutation expression, for example, is associated with intermittent skin solar exposure [11]. Other oncogenes such as CDK4 an CCND1 are also found in solar-exposed skin without mutations in NRAS or BRAF [12].

Based on this understanding, in 2018 the WHO proposed a more comprehensive classification for melanoma [13] that includes clinical, epidemiologic and genetic factors, with additional biomarkers based on these pathological pathways. 

This classification consists of nine categories based on sun exposure:Type I Superficial spreading melanoma with low cumulative sun damage (CSD);Type II Lentigo maligna melanoma;Type III desmoplastic melanoma;Type IV Spitz melanoma;Type V Acral melanoma-related with high CSD;Type VI Mucosal melanoma;Type VII Melanoma in congenital nevus;Type VIII Melanoma in blue nevus;Type IX uveal melanoma, with low or non-CSD.Each category has a specific set of key molecular genes with particular gene mutation amplification or rearrangement of the RNA sequencing.

## 3. Prognostic Factors

It is well known that the presence or absence of estrogen and progesterone receptors and the HER2 protein are used in the basic classification of breast cancer and for predicting treatment and prognosis [14]. In contrast, in the disease scope of melanoma, prognosis does not depend on the tumour’s molecular profile; the main prognostic factor in malignant melanoma is Breslow thickness. Other important prognostic factors, which are part of the standard pathological workup, are ulceration, mitosis rate, surgical margins and Clark level, with the latter decreasing in occurrence over time.

Herein we present select molecules involved in melanoma cell transformation, invasion and metastasis, followed by the suggested role of these molecules in determining patient prognosis.

### 3.1. BRAF Gene

The BRAF gene is a proto-oncogene that belongs to the signal transduction protein kinases and affects cell proliferation, differentiation and migration. Mutations in this gene have been found in several cancer types, while its association with malignant melanoma is one of the most studied. Oncogenic mutations in BRAF are found in approximately 40–60% of patients with cutaneous melanoma [15] and activate the mitogen activated protein (MAP) kinase pathway (the most common mutation is V600E, with V600K following) [16]. The V600E mutation comprises 80% of BRAF-related melanomas and is seen mostly in younger patients in anatomical sites protected from sun exposure [17].

Analysis of BRAF mutation status is already standard procedure in diagnosing melanoma. A large meta-analysis [18] showed poorer prognosis and overall survival in patients harbouring the BRAF mutation. In parallel, the association between BRAF mutation and rapid disease progression allowed for the development of BRAF inhibitors, which are one of the greatest developments in the field of melanoma treatment. BRAF inhibitors have shown efficacy as a monotherapy in patients with unresectable, metastatic melanoma and as adjuvant therapy with stage III melanoma with BRAF mutations. Even greater efficacy was demonstrated using the combination of BRAF and MEK inhibitors [19,20].

Though most patients have homogeneous BRAF, heterogeneity has been established; however, its impact on therapeutic response to BRAF/MEK inhibitors or on patient survival has not been elucidated [21].

### 3.2. NRAS Gene

The NRAS gene encodes the NRAS protein, which is involved primarily in regulating cell division. Similar to the BRAF gene, the NRAS gene is a proto-oncogene, for which gene mutation may contribute to cancer formation in humans. The NRAS mutation is found in 15–20% of melanomas tested [22] and is an inaccurate predictor of disease prognosis [23]. However, the NRAS mutation has a role as an independent predictor, specifically in stage IV melanoma, as its expression predicts shorter survival [24].

### 3.3. KIT Gene

The c-KIT gene is another proto-oncogene; it directs the formation of transmembrane Recetyrosine kinase. KIT signalling plays a role in cell survival, proliferation and differentiation. KIT mutations are identified in 3% of all melanomas, specifically in about 35% of acral and mucosal melanomas [25]. Mutations in KIT almost never occur in conjunction with BRAF and NRAS [26], opening the door to therapeutic trials of KIT inhibitors. In contrast to BRAF and NRAS, KIT mutations do not associate with histological subtypes or tumour stage; they do, however, have a close association with increasing age, acral mucosal subtypes of melanoma, and chronic sun-induced damaged sites [27].

### 3.4. NEUROFIBROMIN 1 (NF1) Gene

The NF1 gene encodes for neurofibromin, which is one of the primary negative regulatory factors of the RAS protein through GTPase activity. NF1 is a tumour suppressor gene. Mutations in this gene lead to a common genetic syndrome characterised by café-au-lait macules, neurofibromas and more. Individuals with NF1 mutations have an increased risk of developing desmoplastic melanoma [28]. NF1-mutated tumours occur in 10% to 15% of melanomas. Melanomas with NF1 mutation typically occur on chronically sun-exposed skin or in older individuals, show a high mutation burden, and do not express BRAF or NRAS mutations [29].

The prognosis of all patients with NF1-mutated melanoma has been reported to be significantly worse than patients with other mutation patterns [30].

### 3.5. Plasma Membrane Calcium-Transporting ATPase 4 (PMCA4)

PMCA4 is an enzyme encoded by the AATP2B4 gene in humans [31]. This enzyme plays a critical role in intracellular calcium homeostasis by removing bivalent calcium ions from eukaryotic cells against large concentration gradients [32]. PMCA4 has an extreme influence on male fertility, with a loss of this enzyme resulting in reduced sperm motility (asthenospermia) and infertility [33]. It also has an important role in several cancer types, including breast, colon and pancreatic cancer [34,35,36]. 

In the context of melanoma, pre-clinical investigations have demonstrated that the plasma membrane calcium ATPase PMCA4 inhibits the migration and metastatic activity of BRAF-mutant melanoma cells [37]. Therefore, the common belief is that PMCA4 downregulation is likely one of the mechanisms that lead to enhanced melanoma cell migration and metastasis [38].

A recently published study [39] analysed the prognostic power of PMCA4 mRNA levels in cutaneous melanoma both at the non-metastatic stage as well as after PD-1 blockade in advanced disease. This study revealed significantly different results for males and females with melanoma; however, there was no specific explanation offered for this effect. Results showed that female patients with high PMCA4 transcript levels had a significantly longer progression-free survival, and that high transcript levels derived from RNA-seq of cutaneous melanoma were associated with a significantly longer overall survival and improved prognosis after PD-1 blockade [40].

## 4. Treatment Response and Prognosis

Before the introduction of checkpoint inhibitors in 2011, the median overall survival (OS) of patients with disseminated melanoma was about nine months. According to current data, the median OS is more than two years [41]. Immunotherapy and targeted biological therapies have changed the landscape of melanomas and other solid tumours. The rising incidence and aggressiveness of malignant melanoma has led to continuous research for new treatment strategies, including both mono and combined therapies.

With an arsenal of proven efficient treatments, it is now time to take a step forward and offer each patient the treatment he/she needs according to the specific disease profile. The main biologically targeted treatment given today is a combination of BRAF and MEK inhibitors, with the main immunotherapy treatment being anti-PD1/CTLA-4. We review recent studies showing the effects of these treatments based on the molecular profile of melanoma.

### 4.1. BRAF and MEK Inhibitors

As previously mentioned, the BRAF mutation is found in approximately 40–60% of melanomas. For patients with non-resectable, metastatic, BRAF-mutant melanomas, BRAF/MEK inhibitor treatment can prolong progression-free survival and overall survival [20]. The combination of BRAF and MEK inhibitors is also effective as an adjuvant therapy for stage III melanoma patients in terms of disease-free survival [30,42,43,44,45,46,47,48,49,50,51,52,53,54,55,56,57,58,59,60,61,62,63,64,65,66].

The NRAS-mutant melanoma is refractory to BRAF inhibitors [43]; still, MEK inhibitor therapy improves progression-free survival and therefore might represent a new treatment option for patients with NRAS-mutant melanoma after the failure of immunotherapy [44].

Data are lacking for c-KIT mutations since the mutation pathways are different and the potential for significant efficacy is low.

### 4.2. Anti-PD1

PD1-inhibitor monotherapy has already been proven as an efficient treatment for unresectable melanoma [45] and as an adjuvant therapy for stage III melanoma [46] regardless of PDL1 expression levels. Compared to other immunotherapies such as anti-CTLA4, anti-PD1 showed better results and fewer side effects in treating melanoma patients [47] as a monotherapy. The greatest therapeutic efficacy, however, was seen when using a combination of anti-CTLA-4 and anti-PD-1 [48]. Anti-PD1/CTLA-4 is even advantageous over BRAF/MEK inhibitors as a first-line treatment for BRAF-mutated melanoma, with a 2-year OS of 72% as compared to 52%, respectively (*p* = 0.0095) [49].

The effectiveness of anti-PD1/CTLA-4 for BRAF-mutant melanoma patients is similar to that of other subtypes of melanoma, even though immunotherapy is not targeted for specific mutations. 

Immunotherapy appears to have a comparable or even greater effect in NRAS-mutant melanoma patients compared to other subtypes [50].

Anti-PD1 efficiency was proven in NF1-mutated melanoma patients, with a better median OS observed in mutated tumours as compared to the wild type [30].

## 5. Novel Therapeutic Strategies in the Field of Melanoma

Oncological treatment strategies are moving toward targeted therapy based on the grouping of tumour subtypes, and in the near future, grouping will be on an individual basis.

Currently, data are lacking regarding individual genome-adapted therapy; however, the impressive outcomes of targeted therapy have accelerated discoveries in the field. While treatments are not based on individuals, patients grouped together based on tumour mutation type may be offered different treatment options. In addition to mutation type, patient gender is also considered in groupings, as this is an important prognostic factor in melanoma and affects treatment choice [51]. Moving forward, we need to use the knowledge we have concerning genomic differences, common mutations and biomarkers to study the systemic influence of specific inhibitory agents.

### 5.1. Gender-Adapted Therapy

Female patents have a survival advantage in the case of melanoma, regardless of other tumour characteristics such as Breslow thickness or ulceration [51]. This leads to the assumption that the genomic differences between males and females may be potentially exploited as a therapeutic target. Differences in melanoma mutation burden associated with gender have already been found [52]. Further gender differences could be related to innate factors such as hormone levels, hormone receptor expression, immune system function and cell apoptosis susceptibility [53]. The connection between immune response and mutation burden explains the superiority of females over males regarding survival outcomes. The more that is revealed about the exact gender-related differences in melanoma, the more treatment may be accurately adapted.

### 5.2. Molecular Targeting Strategies

Six hundred and thirty-five biomarkers are associated with malignant melanoma [54]. There are numerous clinical trials aimed at discovering treatments for specific melanoma mutations. Two novel treatments showing promising results with respect to efficacy are:

**Targeted therapy for c-KIT-mutated melanoma:** The c-KIT gene mutation has been observed in gastrointestinal stromal tumours. Imatinib blocks the receptor tyrosine kinase activity of c-KIT and is a first-line therapy for gastrointestinal stromal tumours (GIST) [55]. Inhibitors of c-KIT, including Nilotinib, Dasatinib and Imatinib, have all been assessed in clinical trials and have delivered disappointing results with minimal activity in c-KIT-mutant melanoma patients. Further studies are required, as the long-term efficacy of c-KIT inhibitors in melanoma patients is uncertain, as well as its efficacy compared to other immunotherapies [56,57,58]. Immunotherapy in combination with c-KIT inhibitors provides hope for future therapeutic responses in mutant melanoma.

**MEK inhibitors for NF1-mutation melanoma:** Although NF1-mutant melanoma patients do not usually express NRAS or BRAF mutations, inhibition of the BRAF, MEK and mTOR pathways may be therapeutic since the NF1 mutation increases RAS/MAPK pathway signalling [29]. The greatest evidence of this achieved so far is related to treatment with MEK inhibitors for NF1-mutant melanoma [59,60]. Although therapeutic response to MEK inhibitors is impressive, a long period of treatment is necessary to achieve this response [61]. Immunotherapy and targeted therapy are both found to be effective for the treatment of NF1-mutant melanoma; however, further studies are required to determine the best treatment protocol with minimal side effects.

The field of targeted therapy is continuously evolving, with existing agents being investigated for new combinations and indications. Immunotherapy is also making continuous progress, and new standards of treatment are determined frequently.

**Immunotherapy as a neoadjuvant therapy:** Currently there are no standardized neo-adjuvant treatment regimens; however, there are several trials showing good pathological response rates, ranging from 50–80%, including OpACIN-neo studies and the PRADO trial, which are both phase II studies. These trials have confirmed the safety and high pathologic response rate (approximately 60%) of neoadjuvant Ipilimumab and Nivolumab. Patients with resectable BRAF-mutant stage III melanoma who received concurrent treatment with Dabrafenib+Trametinib+Pembrolizumab triple combination therapy in a new phase II trial had a high pathological response rate (80%) and a pathological complete response of 50% [62]. 

**Adjuvant therapy:** Anti-PD-1 monotherapy is an adjuvant treatment option for stage III melanoma and stage IV resectable patients. Standardized treatment options include adjuvant nivolumab treatment, which has shown significantly longer relapse-free survival (RFS) than adjuvant Ipilimumab in patients with resected stage IIIB, IIIC and IV resectable melanoma. In the KEYNOTE-054 study [46] assessing patients with stage IIIA, B or C melanoma, adjuvant Pembrolizumab had an improved RFS compared to the placebo. The COMBI-AD [42] trial of adjuvant BRAF/MEK-inhibitor therapy and dabrafenib and trametinib also exhibited improved RFS and overall survival (OS) compared to the placebo. In a new trial, KEYNOTE-716 [63], patients with stage IIB or IIC melanoma received pembrolizumab and, compared to the placebo, treated patients showed reduced risk of recurrence and improved distant metastatic-free survival. Although not yet a standardized treatment, this regimen has recently received FDA approval. 

Immunotherapy is discussed as a neoadjuvant and adjuvant therapy in the current standards of treatment in melanoma, while surgery is still an essential part of the treatment when possible. The new immunotherapy drugs and the development of targeted therapy, with a face towards personalised therapy, bring up the question of the place of surgical treatment in the rapidly changing medical field. Surgical treatment is the gold standard for resectable cases and takes an important part in staging determination, but despite its values, it may lead to complications and morbidity. Therefore, a question about the potential of advanced medical treatments to replace surgical treatment has to be asked, at least for cases of mutilating resection surgeries. 

**Triple therapy for BRAF-mutant patients:** the standard of care for BRAF-mutant melanoma patients is either BRAF/MEK inhibitor therapy or immunotherapy. There are three studies that have assessed anti-PD1+BRAF inhibitors + MEK inhibitors. Firstly, IMspire 150 [64] was the only study with significant results; it showed that Atezolizumab + Cobimetinib improved PFS compared to targeted therapy + placebo. This regimen is not commonly used because usually one needs to begin with BRAF-targeted therapy (Vemurafenib + Cobimetinib) before administering Atezolizumab, which may be complicated. In addition, Atezolizumab is often considered to be inferior compared to anti-PD-1 therapy, even though there is no directly comparative data to support this. The second trial, KEYNOTE-022 [65], showed no significant difference in PFS between triple therapy and targeted therapy. The third trial, COMBI-I [66], also showed no significant support for triple therapy using Sparta-DabTram therapy. There is currently an ongoing phase 3 STARBOARD trial [67] assessing the triple combination of Encorafenib + Binimetinib + Pembrolizumab versus Pembrolizumab alone. This is highly relevant because, as opposed to previous studies, STARBOARD is using front-line anti-PD-1 immunotherapy for comparison, as opposed to BRAF/MEK inhibitors, which currently are rarely used as first-line therapy for melanoma treatment. In general, triple therapy is only considered the best treatment approach in patients with high disease burden that is rapidly progressing.

**New immunotherapy combinations**: A new study, RELATIVITY-047, suggested using combined immunotherapy for unresectable or metastatic melanoma with anti-PD-1 and anti-lymphocyte-activation gene 3 (LAG-3) as active agents [68]. While anti-PD-1 is a known and proven agent, anti-LAG-3 is a relatively new immune checkpoint inhibitor, and its role as an inhibitory target seems promising [69]. RELATIVITY-047 [68] compared Relatlimab (a new checkpoint inhibitor) + nivolumab to nivolumab alone and found a significantly improved PFS in favour of the combination therapy. This trial showed that inhibition of these two immune checkpoints provided a significant benefit compared to anti-PD-1 alone with regards to progression-free survival. Currently there are ongoing studies researching the efficacy of this combination therapy as an adjuvant therapy [70]. There are currently pre-clinical studies supporting the use of T cell immunoglobulin and immunoreceptor tyrosine-based inhibitory motif (ITIM) domain (TIGIT), an inhibitory receptor known as Vibostolimab, in metastatic melanoma [71]. A phase III clinical trial is expected to begin relatively soon to compare TIGIT in combination with anti-PD-1 therapy vs. anti-PD-1 therapy alone as an adjuvant therapy for stage IIB, C, III and IV resectable cutaneous melanoma. In addition, there are current, ongoing trials assessing the addition of the BCL-2 inhibitor navitoclax to the combination of dabrafenib + trametinib [72]. There is also a phase 2 BAMM2 trial examining the addition of hydroxychloroquine to dabrafenib + trametinib in patients with stage IIIC and IV BRAF-positive melanoma [73]. Lastly, the use of ipilimumab/nivolumab + the interleukin 6 (IL-6) receptor blocker tocilizumab in patients with untreated, unresectable advanced or metastatic melanoma has been reported. Results are promising, and the study is ongoing [74]. In a very recent study, the phase I AMBER study presented at the 2022 ASCO Annual Meeting, dual T-cell immunoglobulin and mucin-domain containing-3 (TIM-3) with anti-PD1 showed promising anti-tumour activity. This study utilized Cobolimab, a selective anti-TIM-3 monoclonal antibody [75].

An additional study assessed a TOLL-like receptor (TLR)-9 agonist: vidutolimod. This drug has shown promise in patients as a monotherapy as well as in combination with anti-PD-1 therapy, with a response rate of approximately 20% in both regimens; however, it is more durable when combined with anti-PD-1 therapy [76].

## 6. Conclusions

Understanding the molecular pathways involved in malignant melanoma pathogenesis leads to the development of specific targeted therapies and improves the prognosis of melanoma patients. Current treatment options, however, are subpar, and new therapeutic targets and therapies are critical to optimize treatment outcomes and allow for personalized therapeutic approaches for the best patient care.

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
