# Peer review of "Novel Biomarkers and Therapeutic Targets for Melanoma"

_ijms, 2022, doi:10.3390/ijms231911656_

Round 1
Reviewer 1 Report
This review describes some prognostic biomarkers and novel therapeutics for melanoma to allow for personalized therapeutic approaches for best patient care.
The contents are well summarized, however, the following point should be considered to improve this article.
Minor point
1) Reviewer thinks that this review is well organized.
However, this review is hard to understand.
Reviewer recommend to arrange the characteristics of the biomarker and/or
therapeutics in tabular form.
2) This review is hard to read.
Please insert row before each biomarker and therapy.
Ex) “Prognostic factors” section
determining patient prognosis.
(a line)
BRAF gene
3) The expression style should be unified.
In Bibliography section
Journal name : Full name and abbreviation are mixed.
Ex) [5] Clinical Cancer Research
[26] Clin Cancer Res
The above is an example.
Please unify the expression style through the reference section.
3) There are some extra description.
a) P3, line 11 : TheNRAS → The NRAS
[Please put a space between “The” and “NRAS”]
b) P3, line 16 : The C-KIT gene → The c-KIT gene
[The “c” is lower case like at P5 line 24.]
c) “c” is at the beginning of a sentence. Is “c” lower case OK ?
c-KIT inhibitors → The c-KIT inhibitors (?)
Author Response
Thank you very much for your report.
We have corrected the article according to your notes.
1. We added a row before each biomarker. We did not arrange them in a table since we would like to present each one of them independently. If it is still hard to understand that way, we will consider that again
2. We added a row before each biomarker
3. We unified the reference section
4. These changes were made.
Reviewer 2 Report
Only a few minor corrections:
1)Pag.2 second paragraph before the end: “The V600E mutation compromises 80% of BRAF related melanomas and is seen mostly in younger patients in anatomical protected from sun exposure [17].”
-Compromises should be comprises.
-The second part of the sentence in not entirely clear. Maybe “in” should be deleted.
2)Pag.3 third paragraph: “TheNRAS mutation is found in 15%–20% of melanomas tested [22], and is found to be an inaccurate predictor of disease prognosis [23]. The NRAS mutation status is an independent predictor of shorter survival from the diagnosis of stage IV melanoma [24]”
TheNRAS a space is missing
The meaning of the paragraph is somehow unclear as the two sentences seem to contrast with other. Please clarify it.
Author Response
Thank you very much for your report.
We have corrected the review according to your notes.
1. The word “compromises” was changed to “comprises”, and the sentence was changed too
2. The space was added, and the sentence was changed.